# Surveying pulsating auroras

Eric Grono[1] and Eric Donovan[1]

[1]Department of Physics and Astronomy, University of Calgary, Calgary, Alberta, Canada

**Correspondence:** Eric Grono (emgrono@ucalgary.ca)

**Abstract.** The early morning auroral oval is dominated by pulsating auroras. These auroras have often been discussed as if they are one phenomenon, but they are not. Pulsating auroras are separable based on the extent of their pulsation and structuring into at least three subcategories. This study surveyed 10 years of all-sky camera data to determine the occurrence probability for each type of pulsating aurora in magnetic local time and magnetic latitude. Amorphous Pulsating Aurora (APA) is found to be a nearly ubiquitous early morning aurora, having an 86 % chance of occurrence at its peak. Patchy Pulsating Aurora (PPA) and Patchy Aurora (PA) are less common, peaking at 21 % and 29 %, respectively. Before local midnight, pulsating aurora is almost exclusively APA. Occurrence distributions of APA, PPA, and PA are mapped into the equatorial plane to approximately locate their source regions. The PA and PPA distributions primarily map to locations approximately between 4 and 9 $R_E$, while some APA maps to farther distances, suggesting that the mechanism which structures PPA and PA is constrained to the inner magnetosphere. This is in agreement with Grono and Donovan (2019), which located these auroras relative to the proton aurora.

## 1   Introduction

If one looks at the aurora for just a few hours, it is obvious that there are different types. If one looks at enough aurora, it becomes apparent that a relatively small number of specific auroral types dominate the overall phenomenon. Historically, early auroral researchers classified auroras based on their appearance. This morphological classification lacks any connection to the magnetospheric or magnetosphere-ionosphere-coupling mechanisms that might cause a specific type of aurora.

More recently, auroral types have been considered with regard to the physical drivers of these processes, and great headway has been made differentiating them based on the mechanism responsible for their particle precipitation. In the broadest sense, there are two types of mechanism corresponding to two overarching auroral classifications. In some auroras, electric fields parallel to the magnetic field — so called parallel electric fields — increase particles' kinetic energy parallel to the magnetic field, shifting their pitch angle into the loss cone. Such auroras are classified as *discrete*, an example of which is the auroral arc. In other auroras, stochastic interactions with plasma waves or magnetic field curvature scatter particles' pitch angles into the loss cone. In these cases, the aurora is classified as *diffuse*.

Pulsating auroras are a type of diffuse aurora characterized by quasi-periodic pulsations and precipitating electron energies between a few keV and hundreds of keV (Johnstone, 1978). They generally have an irregular, patchy structure (Royrvik and Davis, 1977) which constantly evolves in time (Shiokawa et al., 2010). The spatial size of pulsating auroral structures has been measured to range from one to hundreds of kilometers across (Royrvik and Davis, 1977). Measurements of pulsating aurora altitude thicknesses are scarce, but Jones et al. (2009) measured a pulsating auroral patch to between 15 and 25 kilometres thick.

Pulsating aurora events occur most often in the morning-sector where they persist for an average of 1.5 hours (Jones et al., 2011; Partamies et al., 2017), but events lasting upwards of 15 hours have been observed (Jones et al., 2013). It is unknown exactly how long pulsating aurora events can persist for. Measurements of pulsating aurora event durations are conservative because ground-based cameras, our primary tool for optically observing the aurora, cannot operate past sunrise (Partamies et al., 2017). The lifetimes of individual structures are known to range from a few seconds to tens of minutes (e.g., Grono et al., 2017; Grono and Donovan, 2018).

Pulsating auroral features exhibit diverse characteristics, varying in terms of shape, size, brightness, altitude, spatial stability, modulation, lifespan, and velocity, yet little effort has been spent on differentiating them. Historically, pulsating aurora was subcategorized by Royrvik and Davis (1977) into patches, arcs, and arc segments, but modern literature generally only refers to "pulsating aurora" and "pulsating auroral patches" (e.g., Yang et al., 2019; Partamies et al., 2019; Ozaki et al., 2019) and would not consider the "streaming arc" of Royrvik and Davis (1977) to be a type of pulsating aurora. Grono and Donovan (2018) recently used all-sky camera data to define criteria for differentiating pulsating aurora based on their phenomenology. They identified three types of pulsating aurora which were separable based on their pulsation and structure. Amorphous Pulsating Aurora (APA) evolves so rapidly in both shape and brightness that it is usually difficult — and often impossible — to uniquely identify structures between successive images at a 3 second cadence. Patchy Pulsating Aurora (PPA) consists of highly structured patches which can persist for tens of minutes and pulsate over much of their area. Patchy Aurora (PA) structures are similar to PPA but do not oscillate in brightness. While it may be oxymoronic to describe a non-pulsating feature as pulsating aurora, PA and PPA are clearly closely related in terms of the underlying scattering mechanism responsible for the precipitation. Based on their appearance in the ionosphere, these two auroras seem to be differentiated only by the existence of a modulating mechanism in the magnetospheric source region. Herein we use the term "pulsating aurora" to collectively refer to APA, PPA, and PA, and the acronyms will be used to identify them individually.

Pulsating aurora has been shown to be pervasive in the morning sector (Jones et al., 2011; Partamies et al., 2017), but we believe those studies conflated at least APA and PPA, while possibly ignoring PA altogether due to its relative lack of pulsation (Grono and Donovan, 2018). Nishimura et al. (2010, 2011) connected specific examples of APA and PPA, without differentiating them, with specific chorus elements in the equatorial magnetosphere. Yang et al. (2015, 2017) related the individual motion of PPA and PA patches to convection in the ionosphere, and their source regions to convection in the magnetosphere. Yang et al. (2019) observed one event where APA was associated with higher energy electron precipitation than was PPA. By locating the latitude boundaries of pulsating auroras relative to the proton aurora, Grono and Donovan (2019) discovered that they occur either within or equatorward of the proton aurora. PPA and PA were observed to occur predominantly equatorward

of the optical b2i (Donovan et al., 2003), which is the ionospheric counterpart to the isotropy boundary for plasma sheet protons and marks the inner boundary of the thin current sheet. APA also occurred there, but in addition, it regularly extended into the transition region where the band of proton aurora luminosity originates from and the magnetic field is stretched.

The aurora is a powerful tool for remote sensing the large-scale dynamics of the magnetosphere. Pulsating aurora is a widespread type of aurora which we do not yet understand the subcategories of. This gap in our collective knowledge limits the information about the state of magnetosphere that can be inferred from pulsating aurora. While surveys of pulsating aurora have been done previously (Jones et al., 2011; Partamies et al., 2017), they have not distinguished between different types. This study presents the first separate surveys of occurrence probabilities for APA, PPA, and PA.

## 2 Data and methodology

To survey pulsating aurora occurrence, the Time History of Events and Macroscale Interactions during Substorms (THEMIS) all-sky imager (ASI) array was used. This network of imagers (Donovan et al., 2006; Mende et al., 2008) is the ground-based component of the NASA mission (Angelopoulos, 2008) designed to study the aurora and substorms using conjoined ground-based and space-borne observations. The ASIs capture panchromatic, or "white light", images of the aurora at a 3 second cadence on a $256 \times 256$ pixel CCD and have been operating for over 10 years, amassing tens of millions of images. It can be surmised from Jones et al. (2011) that pulsating aurora is visible within the order of 10 % of these images (Grono et al., 2017). Of the 21 imagers deployed across North America, those stationed at Rankin Inlet, Nunavut (RANK); Gillam, Manitoba (GILL); and Pinawa, Manitoba (PINA) were utilized for this study. The locations and fields of view of these imagers are shown in Figure 1. The fields of view are drawn at 10 degrees of elevation relative to the imager at an altitude of 110 kilometres.

Data from these ASIs were viewed as keogram-style images (Eather et al., 1976) illustrating the evolution of aurora in time and one spatial dimension, which in this case was aligned to the Gillam magnetic meridian at $-26.1089$ degrees magnetic longitude (MLON). These keograms, an example of which is Figure 2, were arranged and aligned in a stack to give a wide view of the aurora along this meridian so that the upper and lower magnetic latitude boundaries of pulsating auroral events could be identified. The location of this meridian relative to the fields of view of the ASIs can be seen in Figure 1.

Across 2006 through 2016, 280 days were identified where visibility was simultaneously clear at all three sites. Keograms were created for each day of data to search for pulsating aurora. To ensure that the start and end times could be identified precisely and that short periods of pulsating aurora would not be missed, dates were split into multiple 1-hour-sized keograms. However, in general, there was not an integer number of hours of clear data, so each day also had one shorter keogram containing the remaining data.

Within each keogram, the upper and lower latitude boundaries as well as the start and end times of pulsating aurora events were identified by-eye and recorded. One spatial dimension does not necessarily provide enough information to accurately define the boundaries of pulsating aurora, but it provides a reasonable estimate when the alternative is to define the boundaries for hundreds of thousands of individual ASI images. This simple method of defining the event boundaries is often imprecise since the size of the region which pulsating auroras cover can change, in addition to its location. Multiple sets of boundaries

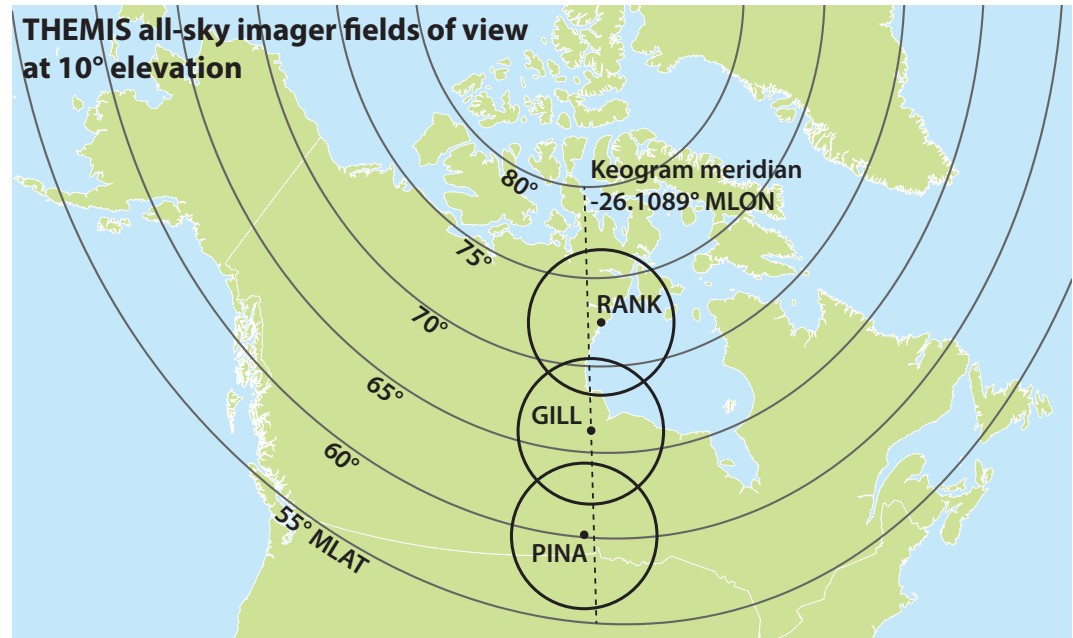

**Figure 1.** The locations and fields of view of the three THEMIS ASIs utilized in this study. The imagers are located in Rankin Inlet, Nunavut; Gillam, Manitoba; and Pinawa, Manitoba. The fields of view are drawn at 10 degrees elevation assuming an altitude of 110 kilometres.

were often used to better define where pulsating aurora was occurring within the keograms in order to compensate. Despite this limitation, more precise and accurate methods of defining the event boundaries are prohibitively time consuming for a dataset
of this size.

The latitude and temporal boundaries were recorded separately for APA, PPA, and PA, which is illustrated in Figure 2. This sample image from the dataset features colour-coded rectangles marking where each pulsating aurora type was identified.

To identify the boundaries of each type, certain characteristic features are searched for. PPA and PA move with ionospheric convection (Yang et al., 2015, 2017; Grono et al., 2017; Grono and Donovan, 2018) and have a stable, well-defined structure
that creates pathlines in keograms. Pathlines trace the trajectory of PPA and PA patches along the keogram meridian and arise due to the long-lived nature of PPA and PA patches. These are the primary signature used for identifying PPA and PA in keograms. Since PA patches do not pulsate, they are separable from PPA by the presence of vertical striations within the pathline which are indicative of pulsation. The appearance of APA within keograms cannot be described as simply, but it is primarily identified by vertical striations and a lack of well-defined structure.

Based on this, it is generally straightforward to uniquely identify each type of pulsating aurora within keograms (Grono and Donovan, 2018), but certain events can be ambiguous and in these instances the full all-sky images were inspected. Occasionally it can be unclear whether an event is APA or PPA. This can be due to a lack spatial information provided by the keogram-style image, but it can also be hard to determine whether an event is APA that is atypically structured, or PPA that is relatively unstructured. Since PPA and PA events will feature many patches, the presence of multiple pathlines can be

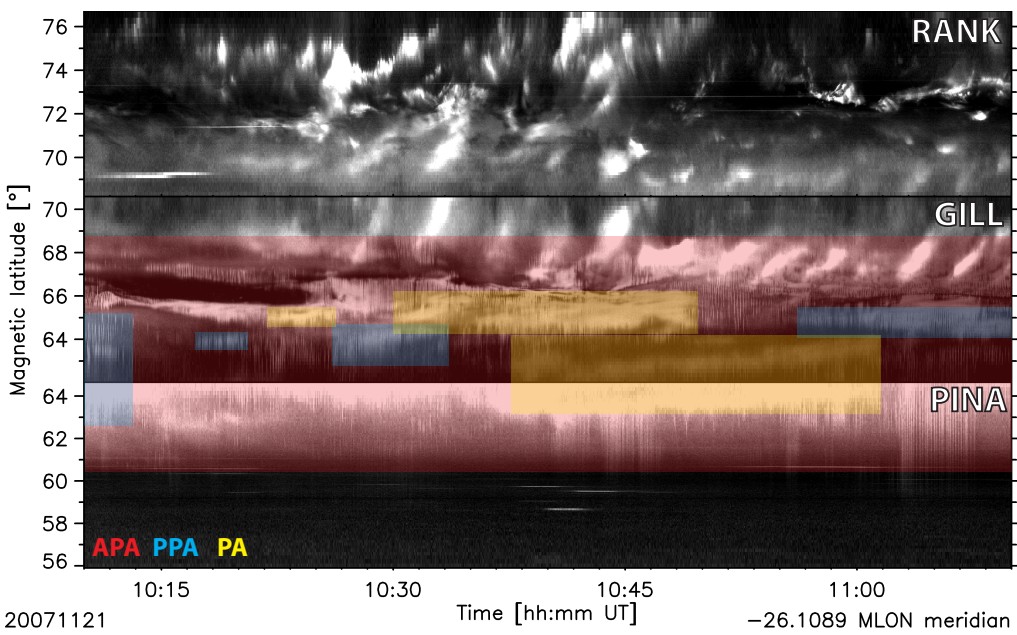

**Figure 2.** An example of a keogram used to approximately define pulsating aurora occurrence. The regions where amorphous pulsating aurora (APA, red), patchy pulsating aurora (PPA, blue), and patchy aurora (PA, yellow) can be identified in the keogram are marked with rectangles. The meridian that the keograms are aligned along is illustrated in Figure 1.

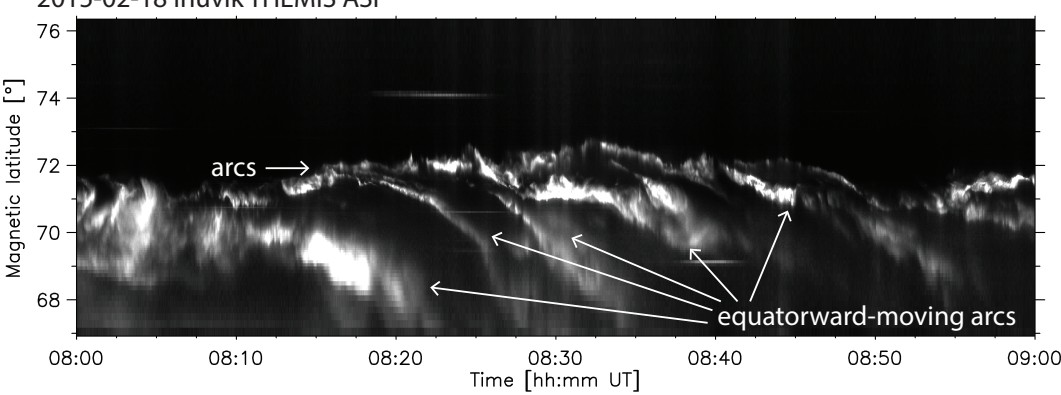

**Figure 3.** An example of equatorward arcs creating multiple pathlines that move in the same direction. These images were captured by the THEMIS ASI stationed in Inuvik, Nunavut on 18 February 2015.

a helpful indicator for recognizing these auroras. PPA and PA move with convection, so the pathlines of multiple patches will have similar trajectories within an event. In addition, APA seems to be present during every pulsating aurora event (Grono and Donovan, 2018), so its presence is a helpful indicator when searching for PPA and PA. The nature of the relationship between these three types of pulsating aurora is unclear, and gradation appears to exist between them (Grono and Donovan, 2018), which can complicate identification.

Pathlines are not exclusive to pulsating auroral patches, however, as arcs can also produce them, as seen in Figure 3. Auroral events featuring multiple equatorward-moving arcs will produce multiple pathlines which move in the same direction. Despite the apparent similarity of such structuring to that of PPA and PA, they are distinguishable from each other with practice.

## 3   Results

We searched 10 years of clear auroral observations, and in Figure 4 we present separate distributions of occurrence probability
for each type of pulsating aurora in magnetic local time (MLT) and magnetic latitude (MLAT). These occurrence probabilities are calculated by dividing the number of days a particular type was observed in a bin by its number of days of clear observations. The number of hours of data of each pulsating aurora type that went into panels 4a–c are 462 hours of APA, 44 hours of PPA, and 58 hours of PA, respectively. These totals are not exact because they are a sum of the timespans each set of boundaries covered, which can overlap. This can be seen in Figure 2, where two PA regions overlap in time. Overlapping boundaries are
not so common as to dominate this calculation and as such these totals are reasonable estimates of the amount of observations. Figure 4d shows the number of days each MLT bin was observed, and within each of these the coverage of the MLAT bins is uniform since our event selection required clear visibility across each ASI.

  APA is seen in Figure 4a occurring in a band from 17 to 7 MLT between 56 and 75 degrees MLAT, peaking during 3.5 to 6 MLT at 66 to 70 degrees MLAT with an ∼86 % probability. This band appears wider at later MLT. Figure 4b shows PPA
predominantly arising in a band from 23 to 6.5 MLT over 57 to 73 degrees MLAT, and to a much lesser extent between 17.5 to 19.5 MLT when the bins are populated by only a single day of data. PPA occurrence probability peaks at ∼21 % from 4 to 5.5 MLT between 65 to 67 degrees MLAT. PA is shown in Figure 4c to occur in a band stretching from 23 to 7 MLT between 59 to 74 degrees MLAT. The peak occurrence probability of PA is ∼29 % between 4 to 5.5 MLT from 65 to 66 degrees MLAT. The latitude of the PA occurrence band is less obviously dependant on magnetic local time than the other pulsating auroras
in these data. This is possibly due to having fewer observations of PPA and PA, which each have on the order 10 % of the observations that APA has. PPA and PA occur in a narrower band than APA. The range of latitudes where pulsating auroras can develop evolves over MLT, following the same trend as the auroral oval, moving to higher latitude with increasing distance from local midnight.

  In Figure 4, the peak activities of each of the three types of pulsating aurora appear to differ in both MLT and MLAT, but they
are difficult to compare when plotted separately as two-dimensional histograms. Figure 5 reduces Figure 4a–c to two separate one-dimensional histograms in MLT and MLAT, allowing the occurrence distributions to be more easily compared. Panel 5a shows that APA covers a larger range of latitudes than PPA and PA, extending farther poleward than both. PPA appears to

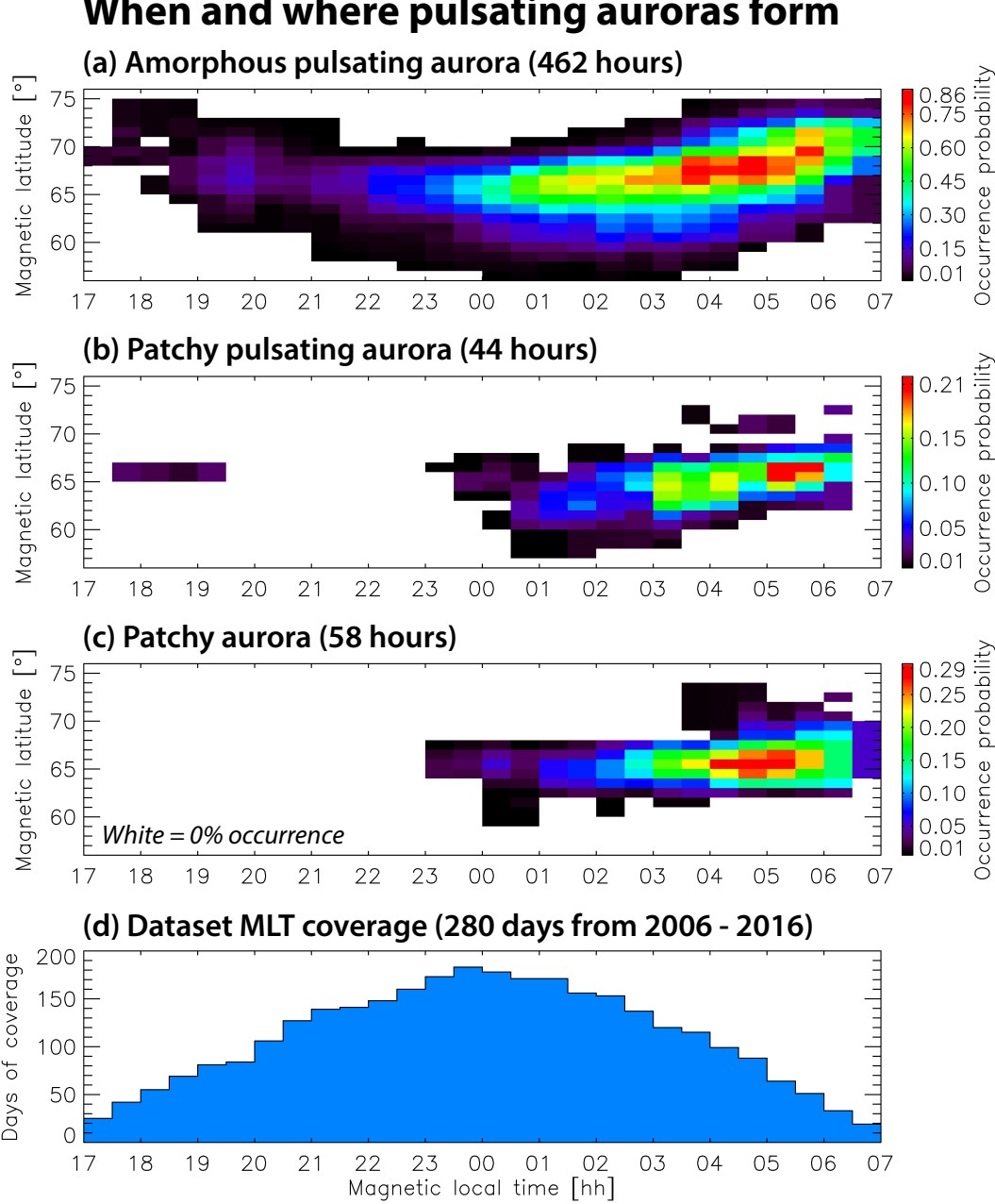

**Figure 4.** Occurrence probability of pulsating auroras based on a survey of times when Rankin Inlet, Gillam, and Pinawa THEMIS ASI had good visibility between 2006 through 2016. White bins in panels (a), (b), and (c) have data coverage but no events, corresponding to a 0 % occurrence probability. Panel (d) shows the number of days of data that had clear visibility in each MLT bin and coverage is uniform across the MLAT bins.

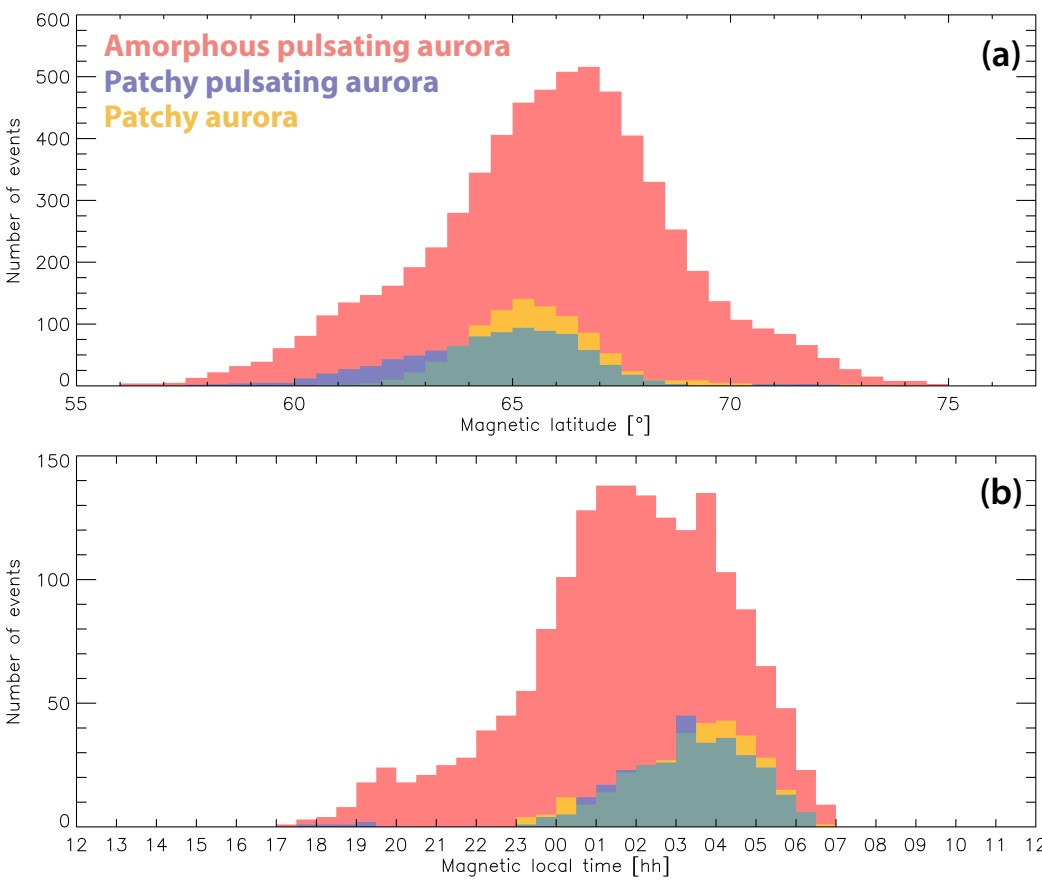

**Figure 5.** Figure 4 reduced in dimension to separate histograms in MLT and MLAT to allow easier comparison of occurrence between the types of pulsating aurora.

develop nearly as far equatorward as APA, although PA does not. Furthermore, the peak occurrence of APA appears to be 1 to 2 degrees MLAT poleward of PPA and PA. In panel 5b, PPA and PA have similar MLT distributions whose peaks approximately align with a local maximum of APA that is ~3 hours later than its peak.

## 4 Discussion and conclusions

The latitude and temporal boundaries of pulsating auroras that were recorded during the survey provide sets of coordinates which can be traced into the equatorial plane of the magnetosphere to estimate the location of their source regions. In this context, a set of boundaries refers to any individual rectangular region used to define the occurrence of pulsating aurora within a keogram, such as those seen in Figure 2. Since a set of boundaries can cover long periods of time and therefore correspond to a large region in the equatorial plane, we split each set into 1 minute slices to more accurately map the shape of its source

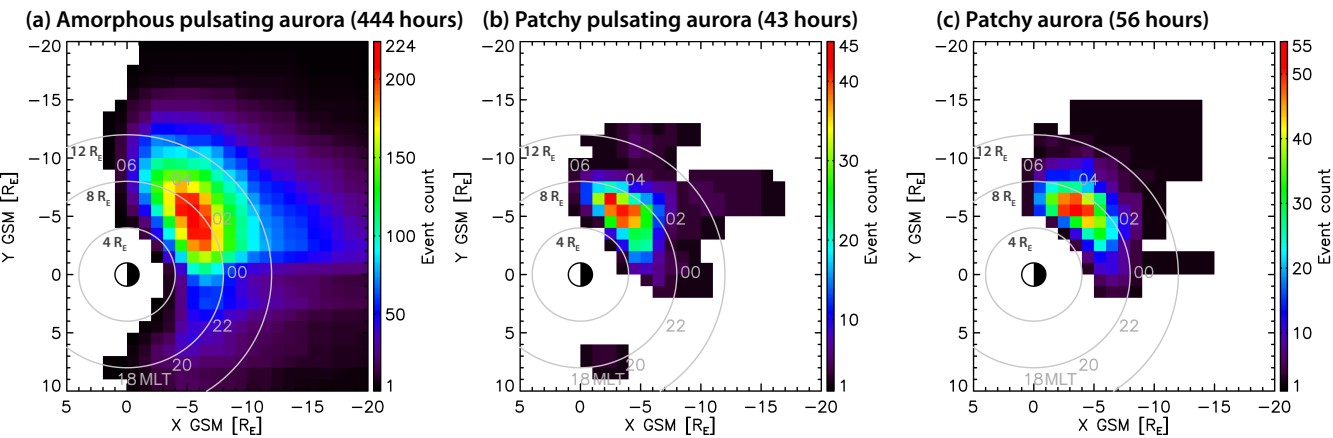

**Figure 6.** Pulsating aurora occurrence mapped to the equatorial plane. Pulsating aurora time and latitude boundaries were mapped using the T89 model (Tsyganenko, 1989) given Kp and the solar wind velocity GSE X component. This figure is based on the same set of events as shown in Figure 4, excluding those with poor solar wind data.

region. The start and end times of each set were rounded down to the nearest minute, and the latitude boundaries were mapped into the equatorial plane at each minute in-between.

Figure 6 shows distributions counting the number of boundaries — that is, the rectangles in Figure 2 — that intersect with
1 $R_E$ by 1 $R_E$ bins when mapped into the magnetosphere using the T89 magnetic model (Tsyganenko, 1989). These bins are in the XY Geocentric Solar Magnetospheric (GSM) equatorial plane where the Z GSM component of the magnetic field changes from being directed away from Earth to toward it. Each boundary is mapped at a one minute time resolution to ensure its shape in the equatorial plane accurately reflects changing geomagnetic conditions.

The T89 model was passed the planetary K-index (Kp) and solar wind velocity Geocentric Solar Ecliptic (GSE) X com-
ponent. Newer models were tested, but the mapped distributions did not meaningfully change. The purpose of mapping the occurrence distributions is to estimate the average location of the source regions, rather than accurately tracing individual events. To this end, the decreased computation time of the T89 model was deemed more valuable than an increase in accuracy which had little impact on the distributions.

Figure 6 does not show a mapping of the occurrence probability, but merely where the events were located. As stated
previously, the purpose of this figure is only to approximately locate the source regions of pulsating auroras. Fewer events are used to create the panels in Figure 6 than Figures 4 and 5 because events were ignored if at least 90 % of the solar wind velocity data was bad data, otherwise the bad data was replaced with values interpolated from the valid data points. According to Figure 6, PPA and PA predominantly originate from a region between roughly 4 and 9 $R_E$. Before local midnight, APA is similarly constrained, however, a portion of the APA distribution maps beyond 9 $R_E$ post-midnight. If you ignore the lowest
population bins, it extends as far out as approximately 15 $R_E$.

These distributions are in agreement with Grono and Donovan (2019), which reported PPA and PA being constrained to more equatorward latitudes than APA relative to the proton aurora. The bright band of auroral luminosity created by proton precipitation is known as "the proton aurora". Proton precipitation occurs when the pitch angles of magnetically trapped protons are scattered as the the particles pass through tight magnetic field curvature in the equatorial plane (Tsyganenko, 1982; Sergeev et al., 1983). The Earthward limit of this stochastic scattering mechanism is the isotropy boundary (e.g., Sergeev et al., 1983), which is located where the magnetic field transitions from being stretched to mostly dipolar. There is an equivalent boundary in the ionosphere, called the optical b2i (Donovan et al., 2003), which marks the rapid decrease of downgoing proton fluxes. Grono and Donovan (2019) found that all pulsating aurora occurred either within or equatorward of the proton aurora. PPA and PA occurred predominantly equatorward of the optical b2i, indicating that they originate from a region where magnetic field topology is mostly dipolar. APA was seen poleward of the optical b2i, but still within the proton aurora.

We know that the proton aurora occurs primarily along and tailward of the transition region between dipolar and stretched magnetic field, and that it occurs at higher latitudes further from local midnight. However, observations of the bright proton aurora's source region have been limited to near magnetic midnight. Spanswick et al. (2017) related the luminosity of the proton aurora to in situ downward proton energy fluxes measured by THEMIS spacecraft in the magnetotail near midnight. They determined that the source region of most proton aurora was located between 6 and 10 $R_E$ at this time, although some could map beyond this. Interpreting this distance range as the farthest limit of pulsating aurora in the equatorial plane near magnetic midnight (Grono and Donovan, 2019), our observations are in agreement.

Global distributions of lower-band whistler-mode chorus (Li et al., 2011), a primary driver of pulsating aurora (e.g., Nishimura et al., 2010, 2011), also indicate that these are realistic distributions of the pulsating aurora source regions. Li et al. (2011) reported lower-band chorus occurrence primarily between 5 and ~8 $R_E$ near magnetic midnight, and a wider occurrence region post-midnight between 5 and 10 $R_E$. They only surveyed events between 5 and 10 $R_E$, the most dominant region for lower-band chorus.

APA is a nearly ubiquitous early morning auroral phenomenon which dominates the morning-sector auroral oval. Pulsating aurora is almost exclusively APA between 17 and 23 MLT, during which time PPA was seen on a single day and PA was never seen. APA is the most common type of pulsating aurora, occurring as often as ~86 % of the time between 3.5 and 6 MLT. PPA and PA occurrences peak at ~21 % between 5 and 6 MLT and ~29 % from 4 to 5.5 MLT, respectively. APA extends farther poleward than PPA and PA, and farther equatorward than PA. The range of latitudes where pulsating aurora can exist varies with MLT, following the auroral oval and reaching its most equatorward latitude at ~2 MLT.

These results are in agreement with recent work by Jones et al. (2011) and Partamies et al. (2017) which reported pulsating aurora occurrence statistics. We suspect that both of these studies included a combination of APA and PPA in their statistics and likely ignored PA. The prevalence of APA indicates that their results should largely reflect the behaviour of APA. Jones et al. (2011) examined 119 days of optical data from the Gillam THEMIS ASI between September 2007 and March 2008, finding that the occurrence rate of pulsating aurora events increases rapidly around magnetic midnight from a small percentage to roughly 50 %. Occurrence increased to nearly 60 % around 3 MLT and remained high until camera shutdown. The APA distribution in Figure 4a is smoother and more strongly peaked than that of Jones et al. (2011), and this is likely attributable to

our larger dataset. A precise comparison between the number of hours is not possible since Jones et al. (2011) only indicated the number of dates their data covered. Compared to our PPA and PA distributions, their occurrence rate is too high and has too early and wide of a peak to likely correspond to a type other than APA. With this consideration in mind, we report a higher peak chance of occurrence than Jones et al. (2011). However, they suggested that their result may be lower relative to past studies by Kvifte and Pettersen (1969) and Oguti et al. (1981) due to the influence of the solar cycle. In contrast, our dataset covers almost an entire solar cycle.

Partamies et al. (2017) surveyed 10 years of optical data from five imagers part of the Magnetometers-Ionospheric Radars-All-sky Cameras Large Experiment (MIRACLE) network of all-sky cameras (Syrjäsuo et al., 1998; Sangalli et al., 2011) between 1997 and 2007. While they did not publish an occurrence probability distribution, they did report that the peak occurrence was between approximately 4 and 7 MLT. Without an occurrence rate to compare to, it is difficult to conclude they predominantly observed APA, but PPA and PA do have a narrower peak in MLT than this. A stipulation of our dataset is that all three ASIs must simultaneously have clear visibility, so our data does not continue past the shutdown time of the lowest latitude camera in Pinawa. The MIRACLE cameras stationed in Lapland that Partamies et al. (2017) analyzed were not similarly constrained and could continue observing later, plausibly explaining why their peak persisted until 7 MLT.

It is unknown which specific mechanisms and conditions are involved in each of these types of pulsating aurora, but structural similarity between PA and PPA (Grono and Donovan, 2018) indicates that they are differentiated only by the existence of modulating processes in the source region. This suggests that pulsation and structuring are the two fundamental aspects of pulsating aurora phenomenology. APA can begin to appear much earlier than PPA and PA, occurrence peaks earlier, and it seems to be the only type that can constitute an entire pulsating auroral event on its own (Grono and Donovan, 2018).

The occurrence distributions of APA, PPA, and PA were mapped into the equatorial plane of the magnetosphere. These mappings correspond to the average locations of their source regions, and they agree with observations reported by other studies. PPA and PA are predominantly constrained between 4 and $\sim$9 $R_E$, while a portion of the APA distribution maps beyond this, as far out as $\sim$15 $R_E$.

Moving forward, there are three key questions pertaining to the conditions and mechanisms driving pulsating auroras: *what processes are responsible for the structuring of PPA and PA?*, *why does PA not pulsate?*, and *does APA play a role in the onset of PPA and PA?*

*Data availability.* The complete set of Figure 2-style images for the entire dataset is available in Grono (2019). THEMIS ASI data is available from http://data.phys.ucalgary.ca/sort_by_project/THEMIS/asi/stream0/. Planetary K-index data was retrieved from the National Oceanic and Atmospheric Administration Space Weather Prediction Center at ftp://ftp.swpc.noaa.gov/pub/indices/old_indices/. Solar wind velocity was obtained via Operating Missions as Nodes on the Internet (OMNI).

*Author contributions.* EG programmed, analyzed, wrote the work, and designed the figures. ED is his supervisor and assisted with analysis.

*Competing interests.* No competing interests are present.

*Acknowledgements.* This research was supported by grants from the Natural Science and Engineering Research Council (NSERC) of Canada and Danish Technical University (DTU). Thanks to Emma Spanswick, Harald Frey, and Stephen Mende for All-Sky data from the NASA Time History of Events and Macroscale Interactions during Substorms (THEMIS) mission.

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
