# Peer review of "Surveying pulsating auroras"

_Annales Geophysicae, 2019_

## Referee Comment (RC1) · Anonymous Referee #1 · 3 Oct 2019

The authors describe a large survey of occurrence rates of recently published sub-categories of pulsating aurora. They classified 10 years of image data from THEMIS network and present the distributions of the different types in magnetic local time and magnetic latitude. They further use the Tsyganenko magnetic field model to map their observations to the equatorial plane to comment on one of the pulsating aurora types having a source region farther toward the tail. The results are new and convincing, the text is clear, rich and understandable. With a few minor changes and a couple of additional discussion items I suggest a prompt publication for this study.

General comments:
– The introduction mentions a recent case study by Yang et al., where it was concluded that APA was related to higher precipitation energies than PPA. Based on the different source regions of APA and PPA found in this study, could you comment on the

agreement of your results with the earlier case study?
– The introduction includes a comment on the previous pulsating aurora surveys probably focussing on a combination on APA and PPA. The discussion could bring this up again to evaluate whether the earlier reported MLT and MLAT distributions agree on the potential type bias of the previous studies.
– Consider adding a map of the locations of the selected THEMIS stations used here.

Specific comments:
Line 80: Were the 1-hour keograms generated for this study, or did they already exist for all THEMIS data? The rest of the sentence seems to say that apart from that one hour, the rest of the day makes a shorter keogram. This is hard to understand.
Line 81: It sounds like the spatial and temporal boundaries of pulsating aurora were detected visually. Could that be explicitly mentioned?
Line 101: The reference on this line seems redundant, as the same thought appears more thoroughly referenced in the previous paragraph.
Line 105: Maybe "pulsating aurora type" instead of "pulsating aurora" ?
Line 116: s the probability here defined as occurrence normalized by the imaging time or nighttime or aurora observation time?

---

## Author Comment (AC1) · 6 Oct 2019

[revised manuscript text omitted]

The authors describe a large survey of occurrence rates of recently published subcategories of pulsating aurora. They classified 10 years of image data from THEMIS network and present the distributions of the different types in magnetic local time and magnetic latitude. They further use the Tsyganenko magnetic field model to map their observations to the equatorial plane to comment on one of the pulsating aurora types having a source region farther toward the tail. The results are new and convincing, the text is clear, rich and understandable. With a few minor changes and a couple of additional discussion items I suggest a prompt publication for this study.

**General comments:**

1. The introduction mentions a recent case study by Yang et al., where it was concluded that APA was related to higher precipitation energies than PPA. Based on the different source regions of APA and PPA found in this study, could you comment on the agreement of your results with the earlier case study?
   a. This paper does not address the particle energies associated with these types of pulsating aurora. The work of Yang et al. (2019) showed one event that suggests precipitating energies may be a distinguishing feature, but it is not firm. We mention their result only to highlight that differences do seem to exist between these classifications of pulsating aurora.
2. The introduction includes a comment on the previous pulsating aurora surveys probably focussing on a combination on APA and PPA. The discussion could bring this up again to evaluate whether the earlier reported MLT and MLAT distributions agree on the potential type bias of the previous studies.
   a. We agree and have included this.
3. Consider adding a map of the locations of the selected THEMIS stations used here.
   a. This has been added as the new Figure 1.

**Specific comments:**

1. **Line 80**: Were the 1-hour keograms generated for this study, or did they already exist for all THEMIS data? The rest of the sentence seems to say that apart from that one hour, the rest of the day makes a shorter keogram. This is hard to understand.
   a. We have expanded on this to make it clearer.
2. **Line 81**: It sounds like the spatial and temporal boundaries of pulsating aurora were detected visually. Could that be explicitly mentioned?
   a. We agree and have added this.
3. **Line 101**: The reference on this line seems redundant, as the same thought appears more thoroughly referenced in the previous paragraph.

      a.   We agree and have removed this reference.

4. **Line 105**: Maybe "pulsating aurora type" instead of "pulsating aurora" ?

      a.   Fixed.

5. **Line 116**: Is the probability here defined as occurrence normalized by the imaging time or nighttime or aurora observation time?

      a.   This has been added to the first paragraph of the Results section.

---

## Referee Comment (RC2) · Anonymous Referee #2 · 21 Nov 2019

This paper describes a large survey of the occurrence of 3 different types of pulsating aurora. The latitudes and local times of the pulsating events are mapped into the equatorial plane using a magnetic field model. This is a valuable addition to the scientific literature and will help to constrain theories for the formation of pulsating aurora and patchy aurora. In general the paper is well written, but there are some parts which are more difficult to follow than they could be.

I recommend that this paper is published in Annales Geophysicae, but have some questions and suggestions for improvements which I have listed below. The line numbers given here refer to the corrected manuscript, "angeo-2019-129-AC1-supplement".

Abstract

Line 4-5: This sentence seems to suggest that pulsating aurora is dominated by the amorphous type at all times (early morning and pre-midnight), without explicitly saying

that. Perhaps more information on the other types could be added to make the meaning clearer and easier to interpret?

Could you add a comment on the relevance of the locations to which the different pulsating aurora types map? Does the location imply something about the source and/or generation mechanism? Maybe refer to the proton aurora here?

1. Introduction

Line 60: The sentence starting "Pulsating aurora is..." could be worded more clearly. How about something like "Pulsating aurora is a widespread type of aurora, but we do not yet understand its subcategories. These subcategories could provide valuable information about the state of the magnetosphere."

2. Data and methodology

Line 84: "would also have" should be "also had" or "also has", or change the "was not" earlier in the sentence.

You state that PPA and PA create pathlines in the keograms. Can you be sure that pathlines created by other features such as arcs could not be mistaken for patches?

Figure 2: What are the features appearing in the RANK keogram north of 72 degrees, that appear to be pulsating? Perhaps it would be helpful to also show a keogram with no pulsating aurora, to contrast with Figure 2?

3. Results

Line 135: "Panel 3a" - I think this should be Panel 4a.

Line 136: "nearly as farther" should be "nearly as far".

Line 137: "panel 3b" should be 4b.

Line 138: The local maximum in the APA MLT histogram seems like it could be a single isolated MLT bin with a larger number of events, but there is actually a slight dip for this

bin in the PA and PPA histograms. Could it be that some PA and PPA was mis-labeled as APA at this time?

In general you don't consider the possibility, likelihood or effect of mis-identification when discussing your results. Is there any way you could quantitatively estimate uncertainties on your occurrence percentages?

4. Discussion and conclusions

Line 149: GSM is used here without the acronym being defined. Although this is a common acronym, it is worth defining for clarity. Similarly for GSE later.

Line 147: I realise explaining the method used is quite complicated, but this paragraph and the following one are difficult to follow. I think you are describing the exact process you use in your computer code, but probably the terminology could be reduced in the paper and the explanation simplified. Instead of using the terms "total bins" and "event bins", can you just say the 1 RE x 1 RE bins shown in Figure 5 count the number of events (i.e. rectangles on the keograms) that intersect that bin, when mapped using T89? Is this correct? You could include the detail that the events are mapped at a 1-minute resolution to determine intersection with the equatorial bins.

Line 158: I think one or two words are missing here around "passed". The sentence doesn't seem complete.

Line 183: "spacecraft" is plural, it doesn't need an s on the end.

Line 185: "This agrees with our observations." - Could you be more specific here? You are not measuring the proton aurora, right?

Line 194 and 195: Is "develop" the right word here? Perhaps "extends" on line 194 and "exist" or "is found" on line 195? To me "develop" implies a location of initial formation, which I don't think is what you mean.

Line 203: Do you mean Figure 4a rather than 3a?

---

## Author Comment (AC2) · 25 Nov 2019

[revised manuscript text omitted]

This paper describes a large survey of the occurrence of 3 different types of pulsating aurora. The latitudes and local times of the pulsating events are mapped into the equatorial plane using a magnetic field model. This is a valuable addition to the scientific literature and will help to constrain theories for the formation of pulsating aurora and patchy aurora. In general the paper is well written, but there are some parts which are more difficult to follow than they could be.

I recommend that this paper is published in Annales Geophysicae, but have some questions and suggestions for improvements which I have listed below. The line numbers given here refer to the corrected manuscript, "angeo-2019-129-AC1-supplement".

**Abstract:**

**Line 4-5:** This sentence seems to suggest that pulsating aurora is dominated by the amorphous type at all times (early morning and pre-midnight), without explicitly saying that. Perhaps more information on the other types could be added to make the meaning clearer and easier to interpret?

*This has been clarified.*

Could you add a comment on the relevance of the locations to which the different pulsating aurora types map? Does the location imply something about the source and/or generation mechanism? Maybe refer to the proton aurora here?

*This has been added.*

**1. Introduction:**

**Line 60:** The sentence starting "Pulsating aurora is..." could be worded more clearly. How about something like "Pulsating aurora is a widespread type of aurora, but we do not yet understand its subcategories. These subcategories could provide valuable information about the state of the magnetosphere."

*Agreed, and fixed.*

**2. Data and methodology:**

**Line 84:** "would also have" should be "also had" or "also has", or change the "was not" earlier in the sentence.

*Agreed, and fixed.*

You state that PPA and PA create pathlines in the keograms. Can you be sure that pathlines created by other features such as arcs could not be mistaken for patches?

*One cannot totally exclude the possibility, but it is generally possible to distinguish them. I identified events using keograms and in many cases where I felt there was ambiguity, I reviewed the full images. In no cases did I find I had confused discrete and diffuse auroras. I find misidentifying the type of pulsating aurora is the larger concern since there seems to be some gradation between the types. The presence of multiple pathlines having similar trajectories during a single event can be a helpful indicator when recognizing PA pathlines. Also, amorphous pulsating aurora (APA) seems to be part of every pulsating aurora event (Grono and Donovan, 2018), another helpful clue. A new figure has been added to help address this.*

**Figure 2:** What are the features appearing in the RANK keogram north of 72 degrees, that appear to be pulsating? Perhaps it would be helpful to also show a keogram with no pulsating aurora, to contrast with Figure 2?

*If I understand correctly which features you are referring to, these seem to be arcs. We have added a new figure to help address this.*

**3. Results:**

**Line 135:** "Panel 3a" - I think this should be Panel 4a.

*Fixed.*

**Line 136:** "nearly as farther" should be "nearly as far".

*Fixed.*

**Line 137**: "panel 3b" should be 4b.

*Fixed.*

**Line 138:** The local maximum in the APA MLT histogram seems like it could be a single isolated MLT bin with a larger number of events, but there is actually a slight dip for this bin in the PA and PPA histograms. Could it be that some PA and PPA was mis-labeled as APA at this time?

*We were careful in our classifications, often reviewing videos of the ASI data to ensure we were accurate. As for the chance of misidentification, I do not see a mechanism which could be responsible for so many mislabelled events at this specific time. This would seem to correspond to tens of days worth of data being mislabelled at this particular MLT.*

*I have added "approximately" to this sentence to reflect the fact that they do not exactly align.*

In general you don't consider the possibility, likelihood or effect of mis-identification when discussing your results. Is there any way you could quantitatively estimate uncertainties on your occurrence percentages?

*We discuss ways in which classifications are ambiguous but cannot define a likelihood for misidentification to occur. We believe it is low due to the regular review of keogram data in movies of the full ASI images.*

**4. Discussion and conclusions:**

**Line 149:** GSM is used here without the acronym being defined. Although this is a common acronym, it is worth defining for clarity. Similarly for GSE later.

*Fixed.*

**Line 147:** I realise explaining the method used is quite complicated, but this paragraph and the following one are difficult to follow. I think you are describing the exact process you use in your computer code, but probably the terminology could be reduced in the paper and the explanation simplified. Instead of using the terms "total bins" and "event bins", can you just say the 1 RE x 1 RE bins shown in Figure 5 count the number of events (i.e. rectangles on the keograms) that intersect that bin, when mapped using T89? Is this correct? You could include the detail that the events are mapped at a 1-minute resolution to determine intersection with the equatorial bins.

*Fixed.*

**Line 158:** I think one or two words are missing here around "passed". The sentence doesn't seem complete.

*Fixed.*

**Line 183:** "spacecraft" is plural, it doesn't need an s on the end.

*Fixed.*

**Line 185:** "This agrees with our observations." - Could you be more specific here? You are not measuring the proton aurora, right?

*Fixed.*

**Line 194 and 195:** Is "develop" the right word here? Perhaps "extends" on line 194 and "exist" or "is found" on line 195? To me "develop" implies a location of initial formation, which I don't think is what you mean.

*Fixed.*

**Line 203:** Do you mean Figure 4a rather than 3a?

*3a is what was intended. This section is comparing the probability of occurrence between the two studies, which Figure 4 does not contain. While Jones et al. (2011) does not feature a latitude distribution and ours is in MLAT and MLT, we think Figure 3 is the more insightful comparison, as opposed to Figure 4b.*